# Selective Laser Melting of Stainless Steel 316L with Face-Centered-Cubic-Based Lattice Structures to Produce Rib Implants

**DOI:** 10.3390/ma14205962

**Published:** 2021-10-11

**Authors:** Cho-Pei Jiang, Alvian Toto Wibisono, Tim Pasang

**Affiliations:** 1Additive Manufacturing Center for Mass Customized Production, National Taipei University of Technology, Taipei 10608, Taiwan; alviantotow@gmail.com; 2Department of Mechanical Engineering, National Taipei University of Technology, Taipei 10608, Taiwan; 3Graduate Institute of Manufacturing Technology, National Taipei University of Technology, Taipei 10608, Taiwan; 4Department of Mechanical and Manufacturing Engineering and Technology, Oregon Institute of Technology, Klamath Falls, OR 97601, USA; tim.pasang@oit.edu

**Keywords:** face-center-cubic, lattice structure, 316L, selective laser melting, mechanical properties

## Abstract

Selective laser melting has a great potential to manufacture biocompatible metal alloy scaffolds or implants with a regulated porosity structure. This study uses five face-centered cubic (FCC) lattice structures, including FCC, FCC-Z, S-FCC, S-FCC-Z, and FCC-XYZ. Specimens with different lattice structures are fabricated using two laser energy densities, 71 J/mm^3^ and 125 J/mm^3^. Density, tensile, compressive and flexural test results exhibit the effect of laser parameters and lattice structure geometries on mechanical properties. The higher laser energy density of 125 J/mm^3^ results in higher properties such as density, strength, and Young’s modulus than the laser energy density of 71 J/mm^3^. The S-FCC lattice has the lowest density among all lattices. The mechanical tests result show specimen with FCC-XYZ lattice structures fabricated using a laser energy density of 125 J/mm^3^ meet the tensile properties requirement for human ribs. This structure also meets the requirement in flexural strength performance, but its stiffness is over that of human ribs. The compression test results of lattices are still incomparable due to unavailable compression data of the human ribs. In short, The FCC-XYZ lattice design fabricated by the 125 J/mm^3^ laser energy density parameter can be used to manufacture customized rib implants.

## 1. Introduction

Thoracic damages from vehicle collision cases are the second most frequent injuries (after injuries to the head), leading to the trauma associated with deaths. Among these thorax bones, ribs incur the highest fracture; thus, surgical rib fixation with implants is commonly adopted [1,2]. The development of implants for ribs has become a critical issue. Implants made from stainless steel, cobalt-based alloys, pure titanium, and titanium-based alloys are commonly selected since they provide certain advantages, such as high mechanical strength, durability, and chemical/biological compatibility over ceramics and polymers. The most commonly used among these alloys is medical grade austenitic stainless steel 316L (SS316L) [3,4]. SS316L offers some benefits due to its good availability, low cost, and good corrosion resistance in various environments. However, the bulk SS316L has much higher density and stiffness than the bone tissue of ribs, which may cause adverse effects such as bone deterioration [5]. Another challenge that may cause the failure of SS316L implant is the potential possibility for secondary phases or precipitation phases in material to occur during implant fabrication or working conditions [6,7]. The precipitates lead to localized corrosion of the SS316L when subjected to human body fluids and release unused elements that may become toxic inside the human body [8,9,10]. Therefore, efforts to fabricate good austenitic stainless steel that matches the density and stiffness of human ribs are necessary. A strategy to achieve these aims is to needs is to modify the solid SS316L into the lattice structured SS316L and ensure that this modified SS316L has no secondary and precipitation phases [11,12,13].

Lattice structures have become attractive prospects for metallic implants because they are lightweight and have desirable mechanical properties. The proper design and fabrication of these structures can reduce the mismatch of the density and stiffness between SS316L and its application as a rib implant. These structures should have similar density, lower stiffness represented by Young’s modulus, and higher strength than natural human ribs. Moreover, the SS316L lattices should not easily corrode in the human body. Thus, the appearance of precipitation phases in the austenite of SS316L should be limited. Zhu et al. reported that the density of the human rib is 0.736 g/cm^3^ [14]. The Young’s modulus of the rib under flexural condition is 2.79 to 7.44 GPa [1]. Feuer et al. reported that the flexural strength of the rib is 38.64 to 80.98 MPa [15]. Michael et al. showed that the human ribs have a Young’s modulus of 10 to 17 GPa, 60 to 100 MPa for their yield strength, and 80 to 120 MPa for their ultimate tensile strength [14,16]. These properties are listed in Table 1. For SS316L lattice structures, their properties are affected by two factors involving fabrication methods and structure geometries.

Benhart et al. introduced a metal lattice structure called metal foam or cellular metal structure produced by conventional metal foaming processes such as bubbling gas through molten metal, sintering hollow sphere metal powder, and depositing metal onto polymer foam [17]. These processes have a limitation of controlling the lattice geometry and its properties. Nowadays, metal additive manufacturing such as direct metal laser sintering (DMLS), electron beam melting (EBM), and selective laser melting (SLM) show the potential to overcome this limitation.

Selective laser melting, a type of laser-based powder bed fusion of metal AM system (ISO/ASTM 52911-1:201), is widely known for its capability to process metals alloys. The SLM can be used to fabricate metal’s complex geometries, such as lattice structures with the interconnected strut and porosity directly from three-dimensional (3D) virtual objects into neatly shaped metal products [18,19,20]. In the SLM process, layers of pre-deposited powders are melted using a high-energy-density laser beam. Some parameters need to be adjusted to provide the high energy density laser beam (J/mm^3^), such as laser power, *P* in (W), average scanning speed, *v* in (mm/s), laser point distance, *d* in (mm), hatch spacing, *h* in (mm), layer thickness, *t* in (mm), and exposure time, *θ* in (s). The relation of laser energy density (*E_v_*) to all of these parameters is represented by Equation (1) [13]:(1)$$E_{v}=\frac{P}{v\cdot h\cdot t},\;v=\frac{d}{\theta }$$

Mustofa et al. found values between 104.2 to 156.3 J/mm^3^ as the critical laser energy density (Ec) for SLM fabrication of SS316L. They described that applying lower laser energy density than the Ec results in a lack of fusion, void formation, balling formation, and unstable melting conditions. On the other hand, implementing a higher laser energy than the Ec promotes spatters and vaporization [21]. Furthermore, this energy also affects the quality of SLM results in terms of microstructure, density, mechanical properties, surface roughness, and final accuracy [18,19,20]. Mustofa et al. mentioned the value of 8 g/cm^3^ as the density of a bulk SLMed SS316L [21]. According to previous works, the SLMed SS316Ls have the ultimate tensile strength ($\sigma_{UTS}$) in the range of 200 to 600 MPa, yield strength ($\sigma_{yt}$) in the range of 150 to 300 MPa [22], and Young’s modulus ($E_{t})$ in the range of 140 to 205 GPa [23]. Jiangwei L et al. observed the homogeneity of the austenite phase under various SLM scanning speeds by XRD tests. The results did not show the occurrence of any precipitation phases, meaning that the chance of corrosion of SS316L SLMed products into the human body is lower [24]. However, the bulk SS316L still has very high density and mechanical properties compared with those of the human rib, listed in Table 1. Therefore, geometries for lattice structures should be investigated further.

In the lattice geometries, the properties of the overall lattice structure are affected by those of their unit cell. A cell, consisting of struts, members, and porous, has a specific shape, configuration, size, and orientation. Olaf et al. defined various cubic-based lattice cells, including body-centered cubic (BCC) and face-centered cubic (FCC). The BCC has body-diagonal struts while the FCC has face-diagonal struts. The body-diagonal struts have an angle of 35.5°, while the face-diagonal struts have an angle of 45°. The modification of both by the addition of vertical z-pillars and resulted in BCC-Z and FCC-Z. The combination of the BCC, FCC, and z-pillars resulted in F2BCC and F2BCC-Z. They were fabricated from SS316L by SLM and designed with struts with a diameter of 0.5 mm and unit cell size of 14.4 × 14.4 × 14.4 mm^3^. Under the uniaxial compression test in the vertical z-direction, their yield strength was in the range of 0.8 to 12.3 MPa [13,25]. Another BCC micro lattice with a strut diameter of 0.2 mm and cell size of 2.5 mm fabricated by Tsopanos et al. had a strength and Young’s modulus for the SLMed BCC micro lattice of 17.6 MPa and 0.44 GPa, respectively [20]. In short, SLM is known for its capability to fabricate SS316L lattices, and the lattice unit cell shapes influence their mechanical properties. However, the structures from previous studies still cannot be compared with the properties required for human ribs.

The work aims to show the possibility of lattice structures of SS316L fabricated by SLM for rib implant application. These structures should fulfill the properties based on their compatibility with human ribs in terms of density, mechanical properties, and chemical stability. This work uses five types of lattice structures and compares their effects on the properties of SS316L. Two lattices, including FCC and FCC-Z, were adopted from Olaf et al.’s work [25] and three lattices, including S-FCC, S-FCC-Z, and FCC-XYZ, were new lattice types. Each unit cell had a 5 × 5 × 5 mm^3^ size and was composed of 1 × 1 mm^2^ struts. The two laser energy densities (71 J/mm^3^ and 125 J/mm^3^) were performed in the SLM process. The 125 J/mm^3^ was adopted from the Mustofa et al. [21], and the 71 J/mm^3^ was chosen as a comparison. Some main properties were determined using density, tensile, compression, and flexural tests. Additional tests such as SEM-EDX and XRD were used as supports to exhibit the surface morphology of the SLM results, their chemical compositions, and their tendency to induce phases instead of austenite.

## 2. Materials and Methods

### 2.1. Materials

The as-received material for this study is SS316L powder (Chung Yo Materials Co., Kaohsiung, Taiwan). Before pouring it into a powder supply chamber, it was dried at 200 °C for 30 min to remove moisture.

### 2.2. Lattice Design

Five types of face-centered cubic (FCC) lattice structures were used. Figure 1 shows the lattice structures of FCC, FCC-Z, S-FCC, S-FCC-Z, and FCC-XYZ.

A lattice was built using struts in the X direction, struts in the Y direction, struts in the Z direction, and diagonal struts. These struts were arranged in the (001), (100), and (010) planes. Every FCC-based lattice had dimensions of 5 × 5 × 5 mm^3^ in length, width, and height. Each strut in the lattice was 1 × 1 mm^2^ in width and thickness and the cubic porosity inside each lattice was 4 × 4 × 4 mm^3^. These dimensions are presented in Figure 2a. Cavities between struts and members allow a porous connection. The description of the five types of FCC-lattices is given in Table 2.

The lattices were used to produce specimens of different geometry. Specimens for lattice morphology characterization used a single lattice, and density specimens used 1 × 1 × 2 lattices. The dimensions of the mechanical testing specimen were determined using ASTM E8 for the tensile test, ASTM E9 for the compression test, and ASTM E209 for the flexural test. The dimensions of mechanical testing specimens are shown in Figure 2b–d.

### 2.3. Lattice Production

A 3D virtual model with different lattice structures was created by using the computer-aided design (CAD) software, (Solidworks 2018 version, Velizi, France) and converted into stereolithography (STL) format. The Materialise Magics 23.1 software was used to manage and bridge the STL data to position it on the base plate, slice the 3D model into many two-dimensional layers with a constant thickness, and adjust the SLM parameters. The fabrication processes of all specimens used an SLM machine (AMP-160, Tongtai Co., Kaohsiung, Taiwan). This study used two levels of laser power: 114 W and 200 W. The laser-focused diameter was 50 µm. The laser energy density was calculated to be 71 J/mm^3^ and 125 J/mm^3^ because the hatch spacing and scanning speed were 88.8 µm and 600 mm/s, respectively. The layer thickness was 30 µm.

A zigzag scanning strategy was used, with the orientation of the zigzag changed to 67.5 degrees for every layer. Nitrogen was pumped into the building chamber at a pressure of 5 kPa to remove the oxygen. Specimens were built when the oxygen content in the building chamber was less than 1% volume. The oxygen content and pressure were in-situ measured by gas sensor in GR1900D advance instrument machine. After the SLM process was complete, the base plate with the SLMed object was extracted from the un-melted powder and cleaned using an air-pressurized spray. The separation process of the SLMed objects and the base plate used a wire-EDM (electrical discharge machining) machine. These procedures were repeated three times to make three repetitions of the specimens.

### 2.4. Characterization of the Lattice Surface Morphology and the Chemical Composition

A field emission scanning electron microscope (FESEM JSM-7610F, Tokyo, Japan) was used to characterize the surface morphology of the SS316L as-received powder and the lattice structure of the fabricated specimens. Lattice images were captured from the top and side. Image J software (Fiji version) worked on analyzing the as-received powder size distribution. The chemical composition of the powder and the lattice structure fabricated using different parameter values were determined using an Energy Disperse X-ray analyzer (EDX, X-max Oxford Instrument, Oxford, UK). An X-ray diffraction (XRD) spectrometer (Bruker D2Phaser, Karlsdorf, Germany), a using radiation of Cu-Kα from 2theta of 100 to 1000, determined the phase of the SS316L as-received powder and fabricated specimens.

### 2.5. Density

Archimedes’ procedure of ASTM B311 was used to measure the density (*D*) as shown in Equation (2):(2)$$D=A/\frac{\left[A-\left(B-C\right)\right]}{E}$$
where *A* (g) is the mass of test specimens in air, *B* (g) is the apparent mass of the test specimen on water, *C* (g) is the mass of the specimen immersed in water, and *E* (g/cm^3^) is the density of water. Equation (3) also is used to calculate the relative density of the lattice structure $\rho_{cal}$ using the volume fraction data [1]:(3)$$\rho_{cal}=\frac{\rho }{\rho_{0}}\times 100\%=\frac{V_{str}}{V_{bulk}}\times 100\%$$
where *ρ* is the density of the lattice structure, $\rho_{0}$ is the density of bulk, $V_{str}$ is the volume of the lattice struts, and $V_{bulk}$ is volume of the bulk metal. Specimens for this study had a lattice of 1 × 1 × 2 with a dimension of 5 × 5 × 10 mm^3^ and were used to measure ρ and $V_{str}$. A solid specimen was also printed to measure $\rho_{0}$ and $V_{bulk}.$

### 2.6. Mechanical Testing

An HT-2402 servo-controlled material testing machine with a capacity of 50 kN (Hung Ta Co., Ltd., Taichung, Taiwan) was used for the tensile, compression, and flexural tests. Figure 2e shows the position of the lattice and the loading direction during the measurement for mechanical properties. For tensile testing, the pulling displacement rate was 1 mm/min.

The portion of the lattice structure for the tensile specimen was in the gauge section between the upper and lower grip sections. This configuration was used to control the breaking location for the tensile specimens. Machine grippers were used to grasp a section of the specimens. The maximum displacement of the machine crosshead was 15 mm. The gauge length, thickness, width, and cross-section area of tensile were 25 mm, 5 mm, 10 mm, and 50 mm^2^, respectively. The tensile load and displacement history were recorded to plot the stress–strain curve and calculate the tensile strength ($\sigma_{UTS}$), yield strength ($\sigma_{yt}$), Young’s modulus ($E_{t}$), and elongation at break ($\varepsilon_{B}$).

The flexural test used a three-point bending method with a maximum mandrel displacement of 5 mm and a displacement rate of 1 mm/min. The reaction force and displacement data were recorded to calculate the strength ($\sigma_{f}$), Young’s modulus ($E_{f}$), and elongation at yield ($\varepsilon_{y}$) as [26]:(4)$$\sigma_{y}=\frac{M\;y}{I}=\frac{3P_{y}\;l_{0}}{2\;w\;t^{2}}$$
(5)$$E_{f}=\frac{\Delta P\;{G_{L}}^{3}}{48\;\Delta d\;I}=\frac{\Delta P\;{l_{0}}^{3}}{4\;\Delta d\;w\;t^{3}}$$
(6)$$\varepsilon_{y}=\frac{6d}{{l_{0}}^{2}}$$
where M (N·m), y (mm), and I (mm^4^) are the bending moment, the centroid mass, and the inertial moment, respectively. The force at the first plateau start added by 0.02 offset, $P_{y}$ (N), the change in the force in the elastic zone ΔP (N), the change in the displacement in the elastic zone Δd (mm), the gauge length or span length, $l_{0}$ (mm), the specimen width w (mm), and the specimen thickness t (mm) were measured. For this study, the span length, thickness, and width of the specimens were 50 mm, 5 mm, and 10 mm, respectively.

The compression test used a maximum pushing mandrel displacement of 15 mm and a displacement rate of 1 mm/min. The yield of compression strength ($\sigma_{y}$) and Young’s modulus ($E_{c}$) were calculated as:(7)$$\sigma_{y}=\frac{P_{y}}{A_{0}}$$
(8)$$E_{c}=\frac{\Delta \sigma }{\Delta \varepsilon }=\frac{\Delta P\;l_{0}}{A_{0}\;\Delta d}$$
where the chosen L/D ratio was 0.8, the rectangular width, D, was 20 mm, the cross-sectional area, $A_{0}$, was 400 mm^2^, and the gauge length, $l_{0}$, was 15 mm.

## 3. Results and Discussion

### 3.1. Lattice Production

The fabricated specimens with the five types of FCC lattice structures are shown in Figure 3a for the compressive test, Figure 3b for the flexural test, and Figure 3c for the tensile test. Figure 3a shows the specimens for SEM-EDX analysis and density measurement. Figure 3b,c show the SLMed specimens for the flexural and tensile test. This study used two levels of laser energy density to fabricate specimens for each test: 71 J/mm^3^ and 125 J/mm^3^. No defects were present on the lattice structures.

### 3.2. Surface Morphology and Chemical Analysis of the Lattice

FESEM-EDX exhibited the morphology and chemical composition of the as-received powder and the SLMed specimens with the lattice structures, as shown in Figure 4. Figure 4a shows that the as-received powder material is spherical. The diameter of the coarse grains is 50 µm, 30 µm for the medium grains, and the fine grains are less than 10 µm. A small amount of fine grains melts with the coarse grains during the fusion melting process. The spherical geometry of the powder reduces the probability of agglomeration and increases flow when the powder is recoated for a new layer. The powder shapes match with the SLM parameters because the layer thickness is 30 µm. Maria et al. mentioned that the layer thickness for SLM processes is commonly 30–50 µm to accommodate powder deposition on the previous layer [19,22]. The percentages of powder, shown in Figure 4b, with a diameter less than 30 µm, were 96% and caused complete layer deposition. The presence of 25% fine powder also played a role in filling the cavity in the medium and coarse powder during the powder recoating process on the platform.

Figure 4c shows a top view of the lattice structure built using a laser energy density of 71 J/mm^3^ with a magnification of 80×. This laser density gives complete fusion. However, hole defects, balling, and partial melting of the powder are seen. Figure 4c shows the smoother surface of the lattice structure that is built using a laser energy density of 125 J/mm^3^. Figure 4d features fewer hole defects and partially melted powder than Figure 4b.

Figure 4e,f show the FCC lattice’s side views that were built using a laser energy density of 71 J/mm^3^ and 125 J/mm^3^. The partially melted powder is seen in both images, but the surface roughness in Figure 4e is greater than that in Figure 4d. Fine grains have a large surface area to volume ratio, so fine grains are likely to fuse at the periphery of the Gaussian energy distribution of the laser beam to produce spherical particles that adhere to the surface. The greater the laser energy, the more significant are the particles on the side surface. A laser energy density of 71 J/mm^3^ produces fewer particles on the surface lattice of the construction axis than a laser energy density of 125 J/mm^3^.

The chemical composition EDX results are listed in Table 3. The results show that the chemical composition of the 316L changes after the SLM processes. The Ni and Fe content decrease but the Cr content increases because their evaporation rates differ in the low-pressure conditions (0.05 atmosphere). Bourgette measured the evaporation rate of 316L alloying elements in temperature of 760 °C to 980 °C and vacuum pressures. He claimed that the evaporation rate of Cr is higher than Fe, and Ni has the lowest evaporation rates among them [27]. Yong et al. defined that evaporation phenomena in SLM create a mass transfer of melted material to the environment [28]. The EDX result confirms that evaporation causes composition change among the three elements. However, the sequent of evaporation rate is different from Bourgette’s work because his experiment was performed in different temperature and pressure conditions (temperature of 980 °C and pressure of 6.5 × 10^−10^ atmosphere) [27]. In this work, the Fe evaporates more than the others, followed by the Ni. The Cr is the most stable among them. The reduction in Fe and Ni content increases the percentages of Cr.

Figure 5 shows the XRD results for the as-received powder and two SLMed objects with an FCC lattice fabricated using laser energy densities of 71 J/mm^3^ and 125 J/mm^3^. The diffraction patterns for the SLMed objects are the same as the pattern for the as-received powder. The XRD peak analysis shows that this SLMed process produces a stable homogenous phase of austenite.

The solidification of SS 316L basically forms a biphasic composed of austenite and delta-ferrite. Furthermore, the other phases, such as the sigma, chromium carbide, and chromium nitride, may occur during equilibrium cooling conditions or long-time heat exposure. The phase heterogeneity in SS 316L promotes the corrosion of the stainless steel [6,7,8]. The corrosion triggers the release of elements such as Fe, Cr, Ni, and other alloy contents. The Fe, Cr, and other main elements are not classified as toxic, while Ni leads to significant pulmonary toxicity [10]. In SLM, the molten SS 316L is cooled rapidly. The sigma, delta-ferrite, chromium carbide, and chromium nitride phases were solved in austenite solid solution. The rapid cooling results in the homogenous phase of austenite and reduces the tendency of corrosion [29].

### 3.3. Density Measurement

Figure 6 shows the density of the built lattice structures. It shows that a laser energy density of 71 J/mm^3^ is less than that for a structure that uses a laser energy density of 125 J/mm^3^. The lower laser energy produces a low-density SLMed object, but the mechanical properties also should be considered [13]. The FE-SEM images show different densities. Specimens fabricated using s laser energy density of 125 J/mm^3^ have more untargeted partially melted powder that diffuses and becomes spatter inside lattice structures. The trapped powder increases the lattice density. Taban et al. reported that the higher energy density increases the density of 316L SLMed objects. The low laser energy density leaves balling, porosity, and cracking defects on the layer surface; therefore, the following new layer will not properly be deposited with the previous layer [19,22].

The density and volume fraction of the lattice is also affected by geometry. An FCC lattice with additional struts in the Z direction has a higher density than a lattice without Z-struts. Therefore, an FCC-Z lattice is denser than an FCC lattice. An S-FCC-Z lattice is denser than an S-FCC lattice. Diagonal struts also increase lattice density, so FCC-XYZ is the densest of these geometries. The density of these lattice structures depends on the volume fraction.

Table 4 shows information about the lattice densities, including the volume fraction, the relative density values, and the percent porosity of the lattice structures. All of the FCC lattice geometries are 60% less dense than the 316L solid geometry. The porosity percentage for the FCC-based lattice is 61 to 73%. In terms of lightweight, The S-FCC lattice is the most excellent among all lattices. The addition of struts (Z-struts, diagonal-struts, and XY-struts), represented by S-FCC-Z, FCC, FCC-Z, and FCC-XYZ, increases lattice density.

The density of human rib is 0.736 g/cm^3^ [14]. Table 4 lists the density of each lattice and shows that the density of FCC lattice structures produced using both laser energy densities (71 J/mm^3^ and 125 J/mm^3^) have too high densities compared with the density of the human rib. Even though all of these lattices have a higher density than the human ribs, the implementation of lattice geometries significantly reduces the density of bulk SS 316L.

### 3.4. Tensile Properties

The tensile properties of FCC lattice structures are shown in Figure 7. Specimens with the FCC structures constructed using a laser energy density of 71 J/mm^3^ have a yield and ultimate tensile strength of 59.48 ± 7.4 MPa and 71.34 ± 7.5 MPa, respectively. FCC-Z has a higher yield and ultimate strength (62.11 ± 7.2 MPa and 78.38 ± 7.3 MPa) than FCC because it features struts in the Z direction. The yield and tensile strength of S-FCC are 36.21 ± 8.1 MPa and 61.49 ± 8.4 MPa, respectively. S-FCC-Z performs better than S-FCC because it has a higher yield strength of 38.26 ± 7.1 MPa and tensile strength of 63.21 ± 7.3 MPa. FCC-XYZ has a yield strength of 66.03 ± 6.8 MPa and tensile strength of 76.99 ± 6.9 MPa. The Young’s Modulus of these FCC-lattices are 1.39 ± 0.2 GPa, 1.42 ± 0.18 GPa, 0.68 ± 0.2 GPa, 0.85 ± 0.19 GPa, and 1.47 ± 0.19 GPa for the FCC, FCC-Z, S-FCC, S-FCC-Z, and FCC-XYZ, respectively. This study shows that the 125 J/mm^3^ laser density parameter gives a higher yield, tensile strength, and Young’s modulus. The FCC has a yield of 61.81 ± 7.5 MPa, a tensile strength of 80.99 ± 7.4 MPa, and Young’s modulus of 1.47 ± 0.19 GPa. The respective yield strength of the FCC-Z, S-FCC, and S-FCC-Z are 80.38 ± 7.4 MPa, 38.62 ± 8.3 MPa, and 72.28 ± 6.9 MPa. The ultimate tensile strength values are 98.73 ± 7.6 MPa, 66.96 ± 8.7 MPa, and 95.97 ± 6.9 MPa. Their Young’s modulus values are 1.71 ± 0.18 GPa, 0.90 ± 0.23 GPa, and 0.98 ± 0.19 GPa. The FCC-XYZ fabricated by 125 J/mm^3^ has the highest yield of 99.97 ± 6.3 MPa, the tensile strength of 113.75 ± 6.5 MPa, and Young’s modulus of 1.84 ± 0.21 GPa. Rehme et al. said that the existence of vertical lattice in Z-direction massively increases the strength in the Z-direction [25]. Therefore, the strength values of FCC-Z and S-FCC-Z are higher than FCC and S-FCC. However, this additional Z-strut is more balance in isotropic strength behavior. Rehme et al. reported that symmetrical structures such as FCC and FCC-XYZ perform more stable in anisotropy loading [25].

The effect of laser energy densities on the lattice stress–strain curves is shown in Figure 8. The use of 125 J/mm^3^, as higher laser energy, results in the higher strength and Young’s modulus, and the lower elongation at break than the use of 71 J/mm^3^. Maria et al. asserted that higher laser energy causes the higher enhancement of the thermal gradient. The higher thermal gradient promotes the higher heat input, better melt uniformity (confirmed in SEM result of Figure 4b,c), faster solidification, and higher cooling rates [30]. Zhe-Chen et al. stated that these factors improve the material strength, hardness, and fatigue performance. However, improper designs combined with over-high laser energy may cause adverse effects such as too high residual stress and the cracking of the SLMed results [31].

Michael et al. showed that the yield strength, ultimate strength, and Young’s modulus values of a rib in the tensile condition are 60–100 MPa, 80–120 MPa, and 10 GPa [16], respectively. Figure 9 shows the data of several values of yield strength, ultimate strength, Young’s modulus, and elongation at break of FCC-based lattices compare with the desired mechanical properties. FCC-Z, S-FCC-Z, and FCC-XYZ, fabricated by 125 J/mm^3^ laser energy density, have a higher strength than that of the human rib, making these three structures suitable in terms of strength. FCC-Z and S-FCC-Z considerable for their combination of strength and low density; however, the FCC-XYZ is preferable because of its higher strength, durability, and anisotropy tension load [25]. All lattices have lower Young’s modulus and stiffness than the human rib and they perform properly because of the low tendency for stress shielding. Lie et al. defined stress shielding as a mismatch phenomenon where higher loads are transferred to the bone introduced by the stiffer implants and are not conducive to the growth of new bone tissues [32]. The FCC-XYZ fabricated by 125 J/mm^3^ is the most suitable among all lattices since it has the closest Young’s modulus to the human rib.

### 3.5. Flexural Properties

Figure 10 shows the stress–strain curves of all lattices resulting from the flexural test. For specimens were produced using a laser density of 71 J/mm^3^. FCC has a yield of flexural strength of 164.3 ± 17.3 MPa and Young’s modulus of 8.1 ± 0.9 GPa. FCC-Z has a yield strength of 179.7 ± 18.6 MPa and Young’s modulus of 8.4 ± 0.7 GPa. The flexural yield strength and Young’s modulus for S-FCC are 13.9 ± 8.4 MPa and 0.63 ± 0.07 GPa, and for S-FCC-Z geometry, the values are 15.9 ± 7.2 MPa and 0.69 ± 0.07 GPa. The flexural strength and Young’s modulus for FCC-XYZ lattice are 209 ± 20.1 MPa and 14.3 ± 1.2 GPa. The higher laser energy density of 125 J/mm^3^ increases the flexural yield strength and Young’s modulus of the lattices and reduces their ductility, represented by strain at the yield point. The differences of stress–strain curves affected by laser energy densities are shown in Figure 11. The yield strength of FCC, FCC-Z, S-FCC, S-FCC-Z, and FCC-XYZ increase to 194.3 ± 18.8 MPa, 200.6 ± 19.7 MPa, 24.1 ± 9.3 MPa, 25.5 ± 9.1 MPa, and 306.9 ± 26.3 MPa, respectively. In flexural tests, S-FCC and S-FCC-Z show two plateaus. The yield points were determined at the beginning of their first plateau added with 0.02 mm/mm stain. FCC, FCC-Z, S-FCC, S-FCC-Z, and FCC-XYZ exhibit an increase in the Young’s modulus to 9.13 ± 0.8 GPa, 9.39 ± 0.9 GPa, 0.85 ± 0.09 GPa, 0.91 ± 0.09 GPa, and 15.2 ± 1.4 GPa.

The existence of vertical lattice in Z-direction increases not only the tensile properties but also the flexural properties in the Z-direction. Therefore, the strength values and Young’s modulus of the Z-supported lattices such as FCC-Z, S-FCC-Z, and FCC-XYZ are higher than those of FCC and S-FCC. In addition, the existence of the diagonal, X, and Y struts in the plane of (001) affect the properties significantly; therefore, the S-FCC and S-FCC-Z have lower strength, lower Young’s modulus, and higher elongation than FCC, FCC-Z, and FCC-XYZ.

The comparisons between the human ribs and lattices flexural properties are shown in Figure 12. Feuer et al. reported that the flexural strength and Young’s modulus for rib are 38.6–80.9 MPa and 2.79–7.44 GPa, respectively [1,15]. FCC, FCC-Z, and FCC-XYZ produced using both laser energy densities are excellent in a yield of flexural strength because they have strength values higher than human ribs. However, Young’s modulus values of the three lattices are over than the human ribs, which may cause a stress shielding problem [32]. The S-FCC and S-FCC-Z have a low potential for stress shielding problems due to their low stiffness or Young’s modulus than the human ribs; however, their flexural strength values are lower than that of the human rib and may cause premature failure due to over mechanical loading conditions.

### 3.6. Compression Properties

The compression properties of FCC lattice structures are shown in Figure 13, Figure 14 and Figure 15. Using a 71 J/mm^3^ laser density gives a compression strength for FCC, FCC-Z, S-FCC, S-FCC-Z, and FCC-XYZ of 74.8 ± 7 MPa, 94.3 ± 9 MPa, 36.6 ± 5 MPa, 56.4 ± 5 MPa, and 107.2 ± 9 MPa, respectively. The compressive strength values of FCC, FCC-Z, S-FCC, S-FCC-Z, and FCC-XYZ lattices fabricated by 125 J/mm^3^ increase to 84.9 ± 8 MPa, 99.7 ± 9 MPa, 57.5 ± 7 MPa, 80.9 ± 8 MPa, and 114.8 ± 9 MPa, respectively. These values are greater than those for lattices that use a 71 J/mm^3^ laser density.

The Young’s modulus values for FCC fabricated by laser densities of 71 J/mm^3^ and 125 J/mm^3^ are 1.23 ± 0.13 GPa and 2.01 ± 0.21 GPa. The Young’s modulus values for FCC-Z lattice produced both laser densities are 1.50 ± 0.14 GPa and 2.1 ± 0.21 GPa, which are higher than FCC due to the existence of Z-struts. S-FCC and S-FCC-Z, manufactured by 71 J/mm^3^ and 125 J/mm^3^ laser energy densities, result in lower Young’s modulus of 0.13 ± 0.15 GPa and 0.94 ± 0.16 GPa for S-FCC and 0.20 ± 0.12 GPa and 1.19 ± 0.13 GPa for S-FCC due to the lack of diagonal struts in the plane of (001). FCC-XYZ lattice geometry has the highest Young’s modulus. The Young’s modulus for FCC-XYZ lattice structures produced using a laser energy density of 71 J/mm^3^ and 125 J/mm^3^ are 1.58 ± 0.16 GPa and 2.39 ± 0.19 GPa, respectively.

The results of the uniaxial compression test show that a laser energy density of 125 J/mm^3^ gives a significantly higher yield of compression strength and Young’s modulus for all FCC lattice structures than for lattices fabricated using a laser energy density of 71 J/mm^3^. Rehme et al. stated that the existence of vertical lattice in Z-direction increases the strength in Z-direction massively, which only suitable for isotropy loading. On the other hand, symmetrical structures such as FCC and FCC-XYZ perform more uniform compression properties in anisotropy loading conditions [25].

Table 5 shows the mechanical properties of FCC-based lattice structures fabricated by the two laser energy densities collected from tensile, flexural and compression tests.

### 3.7. The Prototype of Lattice in Rib Implant Application

Figure 16a shows the geometry of an implant prototype for the seventh segment of an adult male and female ribs at the anterior position. The curvature of this geometry is 4 m^−1^. The cross-sectional shape is oval with a thickness and width of 12.8 mm and 7.44 mm [33,34,35].

The rib implant uses FCC-based lattice for the oval hollow cross-section shape along 75 to 100% rib segment. Figure 16b,c show the designed CAD model and built object. This design was manufactured using a laser energy density of 125 J/mm^3^. The density of the fabricated prototype is 1.44 g/cm^3^, the relative density is 20.6%, and the percentage of porosity is 79.4%. The density of the prototype rib is less than that of the FCC-based lattice structure because the cavity between the oval tube and lattice structures increases the percentage of porosity. Michael et al. reported that for the purpose of osteoconduction, the lattices are ideally covered by the other microarchitecture lattice structures that have suitable porosity for bone tissues to grow [18]. The microarchitecture lattices are not within the scope of this study. Therefore, the combination of the FCC-XYZ, which provides suitable mechanical properties, and micro lattices may result in proper human rib implants.

## 4. Conclusions

The proposed FCC-based lattice structures fabricated using SLM with a laser energy density of 71 J/mm^3^ and 125 J/mm^3^ successfully resulted in several match performances for human rib implants. Different laser energy densities result in different surface morphologies for the lattice structures. After fabrication, there are changes in the chemical composition because the alloy is exposed to high laser energies in a low-pressure atmosphere. This combination results in non-uniform evaporation for the alloy elements. The composition changes, but the single phase of austenite in the alloy is stable. The density and mechanical properties of lattices are affected by the laser energy and geometry. A higher laser energy density results in a higher density and better mechanical properties for the lattice structure. In terms of lattice geometry, FCC-XYZ exhibits the highest strength and density. S-FCC is the lightest and has the lowest strength. FCC and FCC-XYZ produced using a laser energy density of 125 J/mm^3^ are the most biocompatible in terms of mechanical properties. This lattice has an excellent strength comparable to that of a human rib, as demonstrated by the tensile and flexural measurements. The Young’s modulus for these structures is lower in tensile performance but higher in flexural performance than human ribs, so there is less likelihood of stress shielding, which damages lattices in the working environment.

## Figures and Tables

**Figure 1 materials-14-05962-f001:**
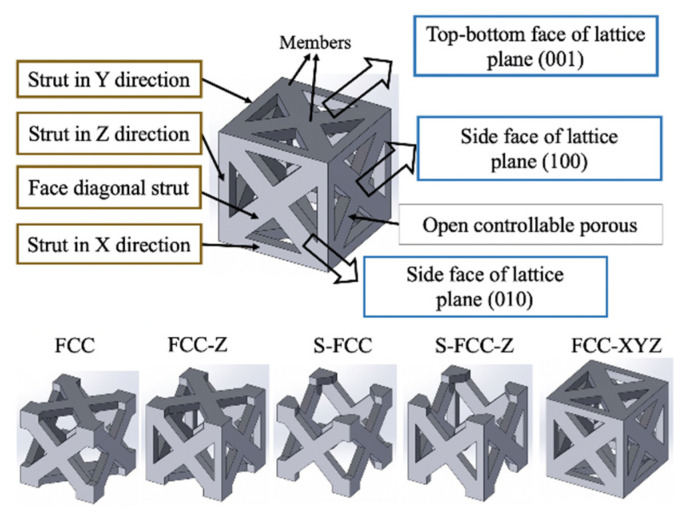
Five types of FCC based lattice structure design.

**Figure 2 materials-14-05962-f002:**
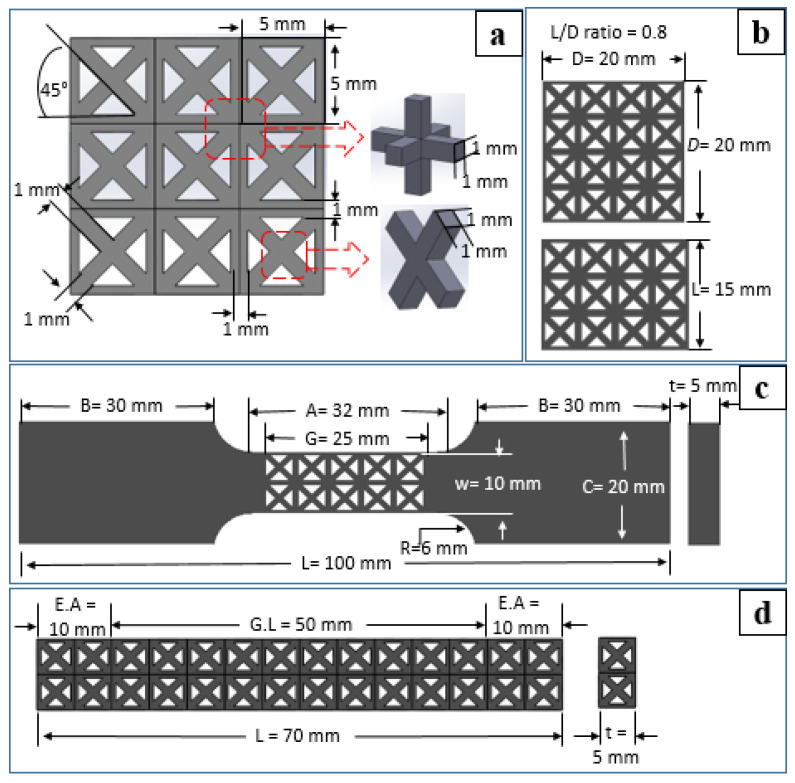
Specimens with lattice structure (**a**) lattice strut’s dimension, (**b**) compression specimen, (**c**) tensile specimen, (**d**) three-point flexural specimen dimensions, and (**e**) loading directions.

**Figure 3 materials-14-05962-f003:**
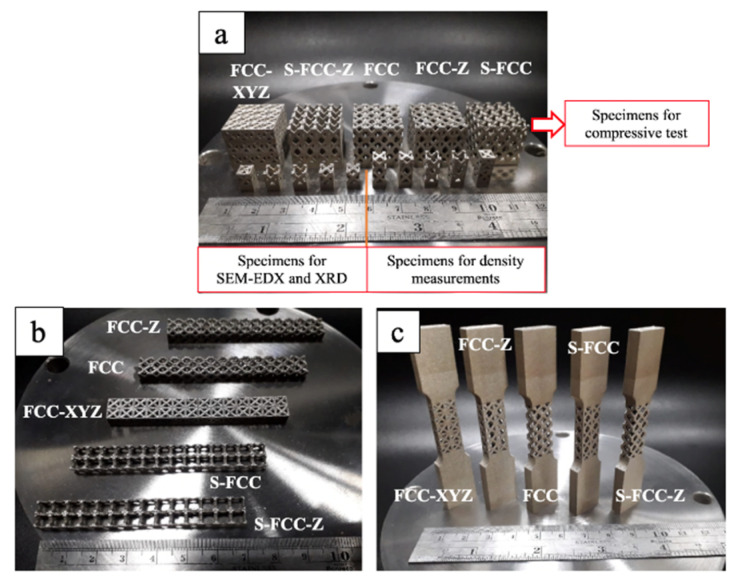
SLM of specimen with five types of lattice structure for (**a**) compression, density, SEM-EDX and XRD tests, (**b**) flexural test and (**c**) tensile test.

**Figure 4 materials-14-05962-f004:**
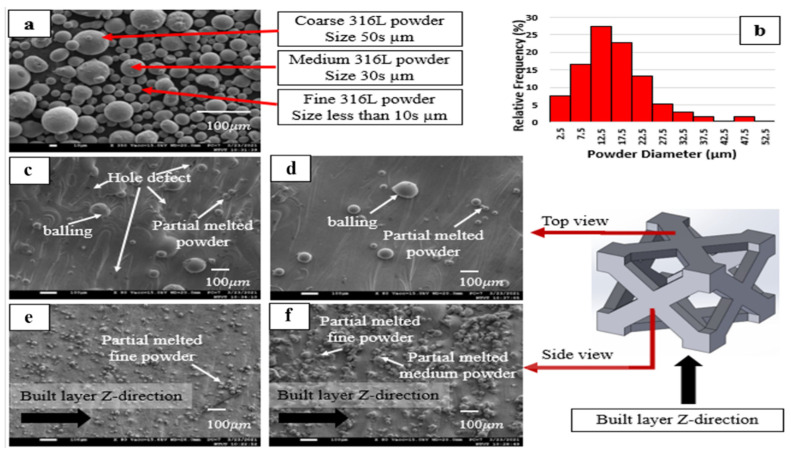
SEM images of (**a**) as-received powder, (**b**) powder size distribution, (**c**) top view of built object by laser energy density of 71 J/mm^3^, (**d**) 125 J/mm^3^, (**e**) side view of built object by laser energy density of 71 J/mm^3^, and (**f**) 125 J/mm^3^.

**Figure 5 materials-14-05962-f005:**
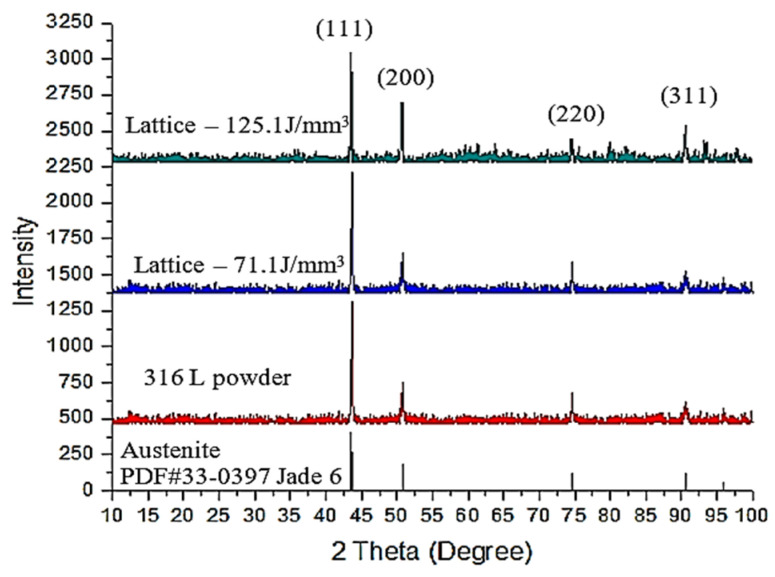
X-ray diffraction patterns of as-received powder built object by laser energy density of 71 J/mm^3^ and 125 J/mm^3^.

**Figure 6 materials-14-05962-f006:**
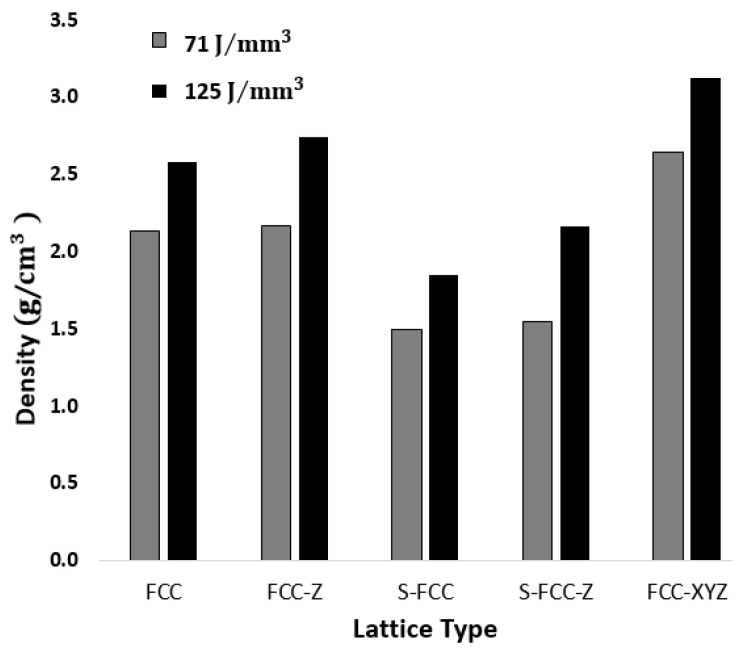
Density of built objects with varying lattice structures.

**Figure 7 materials-14-05962-f007:**
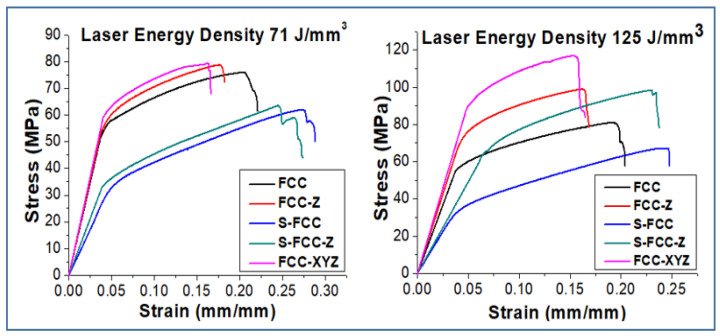
The tensile stress–strain curve of FCC-based lattices affected by lattice geometries.

**Figure 8 materials-14-05962-f008:**
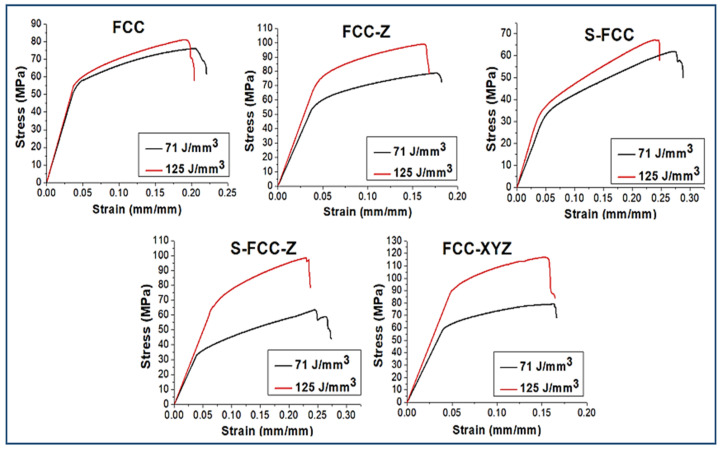
The comparison of tensile stress–strain curve of FCC-based lattices affected by laser energy densities.

**Figure 9 materials-14-05962-f009:**
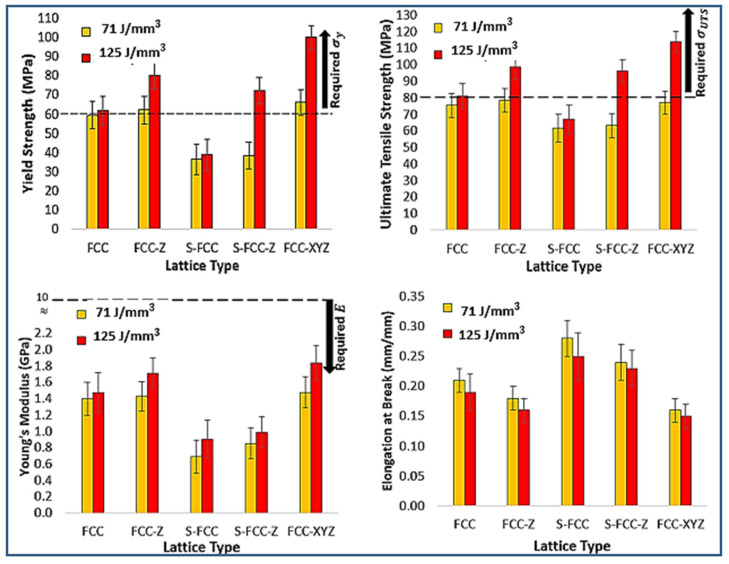
The tensile properties of SLMed objects with five types of lattice structures, yield strength at 0.02 offset from plateau start, ultimate tensile strength, Young’s modulus, and elongation at break.

**Figure 10 materials-14-05962-f010:**
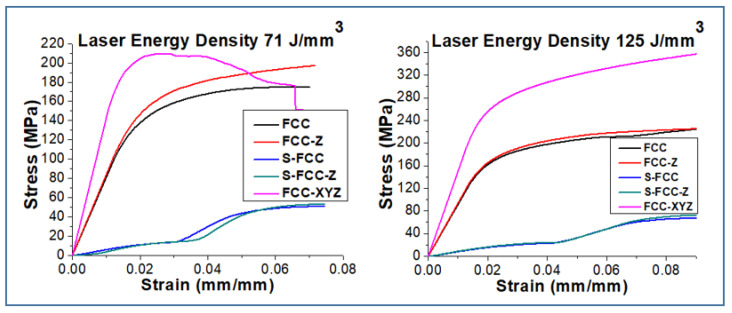
The flexural stress–strain curve from the three-point bending test of FCC-based lattices affected by lattice geometries.

**Figure 11 materials-14-05962-f011:**
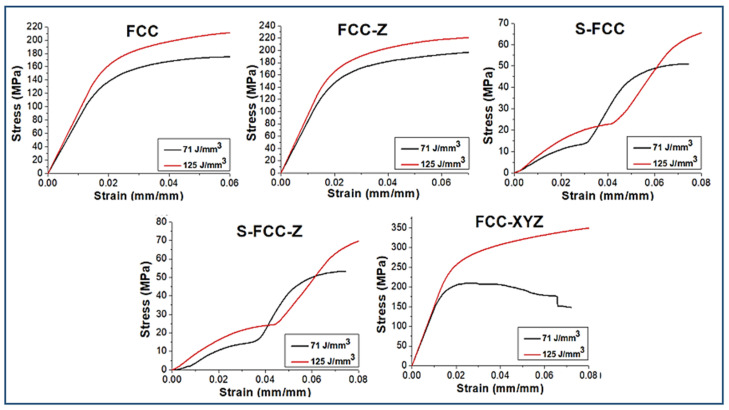
The comparison of flexural stress–strain curve of FCC-based lattices affected by laser energy densities.

**Figure 12 materials-14-05962-f012:**
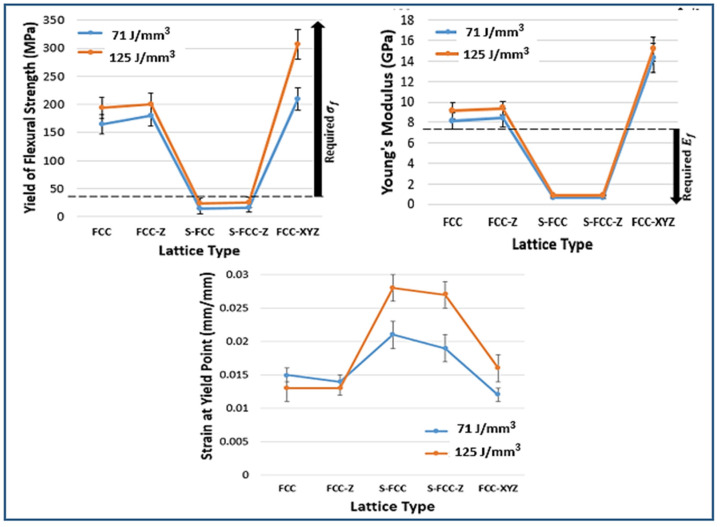
The flexural properties of SLMed objects with five types of lattice structures, yield strength at 0.02 offset from the first plateau start, Young’s modulus, and elongation at yield.

**Figure 13 materials-14-05962-f013:**
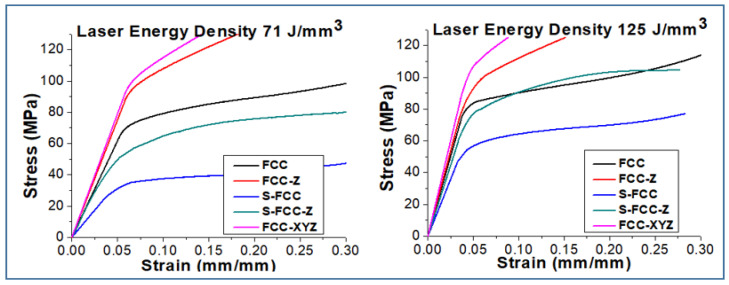
The compression stress–strain curve from the three-point bending test of FCC-based lattices affected by lattice geometries.

**Figure 14 materials-14-05962-f014:**
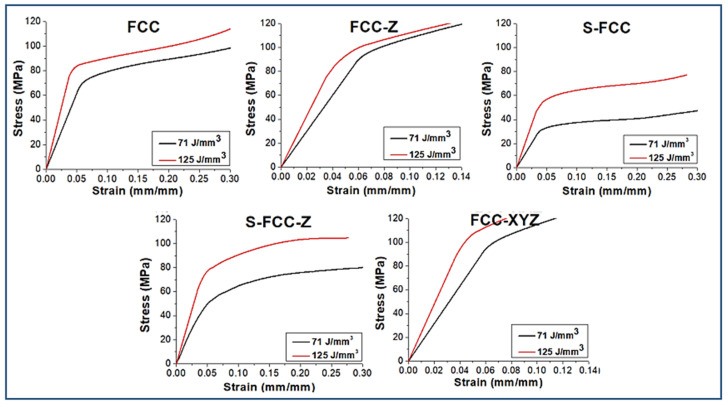
The comparison of flexural stress–strain curve of FCC-based lattices affected by laser energy densities.

**Figure 15 materials-14-05962-f015:**
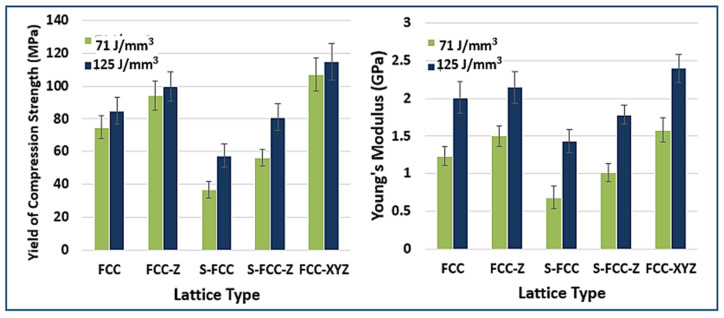
The compression properties of SLMed objects with five types of lattice structures, yield strength at 0.02 offset from the first plateau start, and Young’s modulus.

**Figure 16 materials-14-05962-f016:**
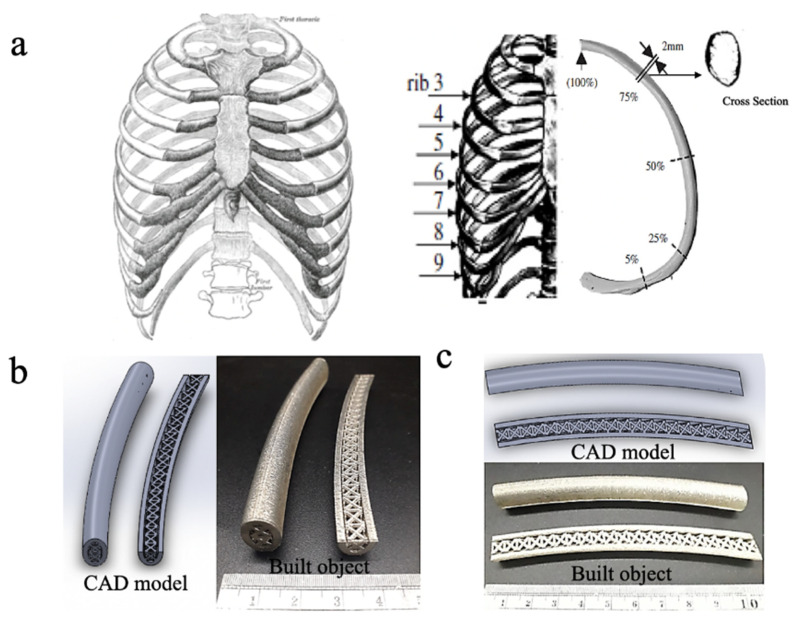
Rib implant images from (**a**) human rib configuration, (**b**) side view and top view (**c**) of CAD model and built object.

**Table 1 materials-14-05962-t001:** The mechanical properties of a human rib.

Human Rib Properties	Values	References
Density (ρ)	0.736 g/cm^3^	[14]
Flexural test		
Young’s modulus ($E_{f}$)	2.79~7.44 GPa	[1]
Strength ($\sigma_{f}$)	38.64~80.98 MPa	[15]
Tensile test		
Young’s modulus (E):	10~17 GPa	[14,16]
Yielding strength ($\sigma_{y}$):	60~100 MPa	[14,16]
Ultimate tensile strength ($\sigma_{UTS}$)	80~120 MPa	[14,16]

**Table 2 materials-14-05962-t002:** Description of the lattices.

Lattice	Descriptions
FCC	Original lattice with diagonal struts arranged on all planes including (100), (010), and (001).
FCC-Z	Lattice with diagonal struts arranged on all planes and addition struts in the Z direction only.
S-FCC	Lattice with diagonal struts arranged on side planes (100) and (010) only
S-FCC-Z	Lattice with diagonal struts arranged on side planes (100) and (010) only and addition struts in the Z direction only.
FCC-XYZ	Lattice with diagonal struts arranged on all planes including (100), (010), and (001) and addition struts in the X, Y, and Z directions.

**Table 3 materials-14-05962-t003:** Chemical composition results from EDX.

Sample	Chemical Composition (wt.%)		
	Cr	Ni	Fe
Powder 316 L	16.05	17.30	66.64
FCC lattice—71.1 J/mm^3^	20.56	14.57	65.07
FCC lattice—125.1 J/mm^3^	19.90	15.00	65.10

**Table 4 materials-14-05962-t004:** Density and porosity of bulk SS316L FCC lattice structures.

Lattice Type	Density (g/cm^3^)		Volume Fraction		Relative Density (%)		Relative Porosity (%)	
	L71 J/mm^3^	L125 J/mm^3^	L71 J/mm^3^	L125 J/mm^3^	L71 J/mm^3^	L125 J/mm^3^	L71 J/mm^3^	L125 J/mm^3^
FCC	2.1	2.6	0.267	0.323	26.7	32.3	73.3	67.7
FCC-Z	2.2	2.7	0.271	0.342	27.1	34.2	72.9	65.8
S-FCC	1.5	1.8	0.187	0.231	18.7	23.1	81.3	76.9
S-FCC-Z	1.6	2.2	0.193	0.271	19.3	27.1	80.7	72.9
FCC-XYZ	2.6	3.1	0.331	0.391	33.1	39.1	66.9	60.9
Solid	8		V_bulk_	1	100		0	

Density of SLMed bulk SS316L without porosity is 8 g/cm^3^ [21].

**Table 5 materials-14-05962-t005:** The mechanical properties of lattice structures.

LED	Lattice Type	Tensile Properties				Flexural Properties			Compression Properties		
		$\sigma_{y}$ (Mpa)	$\sigma_{UTS}$ (Mpa)	E (Mpa)	$\varepsilon_{B}$ (%)	$\sigma_{y}$ (Mpa)	E (GPa)	$\varepsilon_{y}$ (%)	$\sigma_{y}$ (Mpa)	E (GPa)	$\varepsilon_{y}$ (%)
71 J/mm^3^	FCC	59.5 ± 7	75.3 ± 8	1.39 ± 0.26	21 ± 2	166 ± 17	8.09 ± 0.8	1.5 ± 0.1	74.8 ±7	1.2 ± 0.1	7.4 ± 1
	FCC-Z	62.1 ± 7	78.3 ± 7	1.42 ± 0.18	18 ± 2	180 ± 19	8.39 ± 0.7	1.4 ± 0.1	94.3 ±9	1.5 ± 0.1	7.6 ± 1
	S-FCC	36.2 ± 8	61.5 ± 8	0.68 ± 0.22	28 ± 3	14 ± 9	0.63 ± 0.1	2.1 ± 0.2	36.6 ±5	0.7 ± 0.2	8.5 ± 2
	S-FCC-Z	38.3 ± 7	63.2 ± 7	0.85 ± 0.21	24 ± 3	16 ± 7	0.69 ± 0.1	1.9 ± 0.2	56.4 ±5	1.0 ± 0.1	8.3 ± 2
	FCC-XYZ	66.1 ± 7	76.9 ± 6	1.48 ± 0.19	16 ± 2	209 ± 21	14.29 ± 1.2	1.2 ± 0.1	107 ± 9	1.5 ± 0.2	8.1 ± 1
125 J/mm^3^	FCC	61.8 ± 7	80.9 ± 7	1.47 ± 0.18	19 ± 3	194 ± 19	9.13 ± 0.8	1.3 ± 0.2	84.9 ± 8	2.0 ± 0.2	5.3 ± 1
	FCC-Z	80.4 ± 7	98.7 ± 7	1.71 ± 0.17	16 ± 2	201 ± 18	9.39 ± 0.9	1.3 ± 0.1	99.7 ± 9	2.1 ± 0.2	5.1 ± 1
	S-FCC	38.6 ± 8	66.9 ± 9	0.90 ± 0.23	25 ± 4	24 ± 9	0.85 ± 0.09	2.8 ± 0.2	57.5 ± 7	1.4 ± 0.2	6.4 ± 3
	S-FCC-Z	72.3 ± 7	95.7 ± 7	0.99 ± 0.19	23 ± 3	25 ± 9	0.91 ± 0.09	2.8 ± 0.3	80.9 ± 8	1.8 ± 0.1	6.2 ± 3
	FCC-XYZ	99.9 ± 6	114 ± 6	1.84 ± 0.20	15 ± 2	307 ± 26	15.16 ± 1.4	1.6 ± 0.2	115 ± 9	2.4 ± 0.2	6.1 ± 1

## Data Availability

The data presented in this study are available on request from the corresponding author.

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
