# Peer review of "Selective Laser Melting of Stainless Steel 316L with Face-Centered-Cubic-Based Lattice Structures to Produce Rib Implants"

_materials, 2021, doi:10.3390/ma14205962_

Round 1

Reviewer 1 Report

The authors successfully processed different lattice geometry of 316SS parts by using 2 different laser energy densities. Several mechanical properties are compared for different lattice geometries and laser energy densities. The paper might be accepted after following the major revisions below:

1- Line 87: A discussion is done with the study of Mustofa et al. [21] to define the critical energy density in processing 316LL SS powders. Authors later stated that they chose an energy density of 125 J/mm³ to process powders, but there is still a lack of knowledge on choosing 125 J/mm³. Also, the authors cited Mustofa [21] and wrote that minium critical laser energy density is 104.2J/mm³ to avoid lack of fusion, void formation, balling formation, and unstable melting conditions in PBF-LB of 316L SS. Here, we need another discussion of why authors have precisely chosen 71J/mm³ to compare the parts processed with 125J/mm³.

2- Authors stated that SS316L powder was dried at 2000C. I assume it is 200°C. A correction is needed in line 136.

3- The authors stated in lines 174-175 that oxygen content is 1%. Is it wt., vol.%, or? Moreover, it is good to state how oxygen content is measured.

4- In line 280, it is written that EDX results are listed in Table 3. Table 3 is missing in the manuscript.

5- In lines 281-282, the authors stated that the Ni content increased, but the Fe and Cr content decreased. Table 4 shows the opposite as Ni and Fe content decreases and Cr content increases after processing. In addition, the authors stated that Fe evaporates more than others, but the decrease in %Ni content is higher than Fe in Table 4.

6- In Figure 5, XRD patterns are given for Lattice-125.1 J/mm³, lattice-71.1 J/mm³, and 316L powder. The authors matched the patterns with the austenite phase. 316L and Lattice-71.1.J/mm³ have the same diffraction patterns of austenite, but Lattice 125.1J/mm³ has some additional peaks compared to austenite which can be seen between 60-70°, 80-85° and 93°. It is essential to define these peaks and make a comparison with the Lattice-71.1 J/mm³.

7- The authors stated that FCC-XYZ fabricated by 125J/mm³ has the closest Young`s modulus to the human rib. I recommend a discussion on why other build geometries have lower Young`s modulus. A correlation between laser energy density, lattice geometry, and young`s modulus will be helpful for readers.

8- Line 468: Please delete “25”.

9- The resolution of Fig. 15 should be increased.

Reviewer 2 Report

The research has a high interdisciplinarity level with great applicability in medicine field. Based on current references, the state-of-the art is appropriately developed and the progress beyond the state-of-the-art is clearly demonstrated. The research methodology is well presented and the results are adequately discussed. The conclusions are appropriately presented and they are supported by the results. In my opinion, the paper can be published in this form.

Author Response

Thanks and appreciated.

Reviewer 3 Report

Article  Selective Laser Melting of Stainless Steel 316L with Face-Centered-Cubic-Based Lattice Structures to Produce Rib Implants

In general, the paper is topical and very interesting. Also the authors have a decent way of writing a scientific text.

However, there are some serious flaws in the article.

Five types of lattice structures are presented and tested. However, the article does not include a computational analysis of the structures presented. With modern software, FEM analysis is easy to perform and the quality of the structures is already known at this stage. Nowadays it should be done that structures are first optimized by FEM calculation and then the results can be verified by experimental methods.

Printing parameters should be optimised to suit the application. The fact that results are presented with two heat inputs does not provide any new information.

There are several terms for the same metal AM method, selective laser melting one of these. I would recommend that scientific articles use the naming convention recommended by ISO/ASTM 52911-1:201 : Laser-based powder bed fusion of metals (PBF-LB/M).

Round 2

Reviewer 1 Report

I found the study useful for AM to understand the effect of different lattice structures on the physical and mechanical properties of as-built 316L implants.

Reviewer 3 Report

Thank you for the changes. The article is now ready for publication.